# Neural Theorem Proving: Generating and Structuring Proofs for Formal Verification

**Balaji Rao**                                                          BRAO@STEVENS.EDU
**William Eiers**                                                      WEIERS@STEVENS.EDU
**Carlo Lipizzi**                                                      CLIPIZZI@STEVENS.EDU
*1 Castle Point Terrace, Hoboken, NJ 07030*

**Editors:** Leilani H. Gilpin, Eleonora Giunchiglia, Pascal Hitzler, and Emile van Krieken

## Abstract

Formally verifying properties of software code has been a highly desirable task, especially with the emergence of LLM-generated code. In the same vein, they provide an interesting avenue for the exploration of formal verification and mechanistic interpretability. Since the introduction of code-specific models, despite their successes in generating code in Lean4 and Isabelle, the task of generalized theorem proving still remains far from being fully solved and will be a benchmark for reasoning capability in LLMs. In this work, we introduce a framework that generates whole proofs in a formal language to be used within systems that utilize the power of built-in tactics and off-the-shelf automated theorem provers. Our framework includes 3 components: generating natural language statements of the code to be verified, an LLM that generates formal proofs for the given statement, and a module employing heuristics for building the final proof. To train the LLM, we employ a 2-stage fine-tuning process, where we first use SFT-based training to enable the model to generate syntactically correct Isabelle code and then RL-based training that encourages the model to generate proofs verified by a theorem prover. We validate our framework using the miniF2F-test benchmark and the Isabelle proof assistant and design a use case to verify the correctness of the AWS S3 bucket access policy code. We also curate a dataset based on the $\text{FVEL}_{\text{ER}}$ dataset for future training tasks[1].

## 1. Introduction

Recent advances in language models has revolutionized the approach to mathematical reasoning in artificial intelligence. Language models, and in particular large language models (LLMs), have made significant advances in the field of general theorem proving (Guo et al., 2025; Azerbayev et al., 2023). Consequently, formal theorem proving using large language models has recently garnered renewed attention. Formal theorem proving lies at the intersection of mathematics and computer science, where mathematical statements modeling the interaction of computer systems are derived and translated into a formal language which is used to prove the correctness of programs. While formal theorem proving is effective at producing high quality code that provides correctness (Bibel, 2013) it is often a laborious process and requires an intimate level of domain expertise in order to correctly model the computer program into a mathematical statement. Moreover, the costs associated with manual verification can be prohibitively expensive and potentially yield unwieldy proofs which are far more complex than the code being verified.

There have been a number of approaches leveraging machine learning in automated theorem proving which focus on tasks such as premise selection (Irving et al., 2016) and proof search (Loos et al., 2017). More recently, with advancements in LLMs' mathematical reasoning abilities (Azerbayev et al., 2023; Shao et al., 2024), there have been focused efforts on using LLMs for automated proof synthesis. Two main paradigms towards automated theorem proving have emerged: one that generates the whole proof and the other that generates only the next proof step. However, both systems suffer from several shortcomings. Whole proof generators

---

1. The code, dataset and training scripts are available at: `https://github.com/kings-crown/ProofSeek`

are incapable of utilizing previously proved lemmas. On the other hand, proof step generators are not as scalable during training and inference (Xin et al., 2024).

While there are a number of studies that attempt to address the shortcoming of each of these paradigms (Wang et al., 2023; Zheng et al., 2023; Dong et al., 2024), they all concentrate on silos of mathematical processes and build systems to perform better on benchmarks. They do not look at systems built for purposes beyond the mathematical community.

In this paper, we introduce a framework for generalized theorem proving to bridge the gap left by existing neural theorem proving approaches which mainly focus on achieving high proof success rate on standardized benchmarks. Our framework consists of three core modules: the first generates natural language statements of the code, policy, or statement to be verified; the second trains an LLM which generates formal proofs from the natural language statement; the final module employs heuristics from the ProofAug approach (Liu et al., 2025) for building the final proof which can be checked by the proof assistants such as Isabelle. We employ a two-stage fine-tuning process to train the LLM, where we leverage SFT-based training to enable the model to generate syntactically valid Isabelle code, and RL-based training to encourage the model to generate semantically valid Isabelle code. The main contributions of this work are as follows:

- We introduce a framework for generalized theorem proving that enables the verification of natural language statements, including code and security policies.
- We curate a dataset to fine-tune language models for formal theorem proving, improving their reasoning capabilities across diverse domains.
- Our fine-tuned model, PROOFSEEK, outperforms DeepSeek on an unseen problem domain, achieving a 3% improvement in proof success rate while demonstrating a 20% reduction in execution time
- We successfully apply our framework to verify the correctness of AWS S3 bucket policies, showcasing its potential for automated theorem proving in practical applications.

## 2. Background and Related Work

**Formal mathematics and verification**  Formal mathematics is the practice of expressing mathematical statements, proofs, and reasoning in a rigorous language that can be verified for correctness by a computer (Polu et al., 2022). This approach is fundamental to formal verification, which ensures the accuracy of both mathematical proofs and complex systems, including software and hardware (Avigad, 2010). Interactive theorem provers (ITPs) such as Isabelle, Coq, Lean, and HOL Light assist with formalizations by allowing users to encode proofs in a formal language and automatically verify their correctness.

**Autoformalization**  Autoformalization is the process of automatically translating from natural language statements and mathematics to formal specifications and proofs and has gained significant attention in recent years (Wu et al., 2022). Progress in autoformalization systems accelerate the development of tools for mathematical reasoning which can be used in machine learning without the need for associated ground-truths (Wu et al., 2022; Polu et al., 2022). Large language models (LLMs) have shown promising results in this area, demonstrating the ability to translate mathematical competition problems into formal specifications in systems like Isabelle/HOL (Jiang et al., 2022a; Wang et al., 2023; Liu et al., 2025). There are strong argument that autoformalization is a promising path for systems to learn sophisticated, general purpose reasoning in all domains of mathematics and computer science (Szegedy, 2020).

**Automated theorem proving for proof assistants**  Tools like Sledgehammer (Böhme and Nipkow, 2010) automate reasoning within the interactive theorem prover Isabelle by translating goals into other types of logic, which are then sent to automated theorem provers like Z3 (De Moura and Bjørner, 2008) and Vampire (Riazanov and Voronkov, 2001). If they find a proof, Sledgehammer reconstructs it in an applicable format (Zhao et al., 2024). PISA (Portal

to ISAbelle) (Jiang et al., 2021) supports automated proof search for Isabelle and can be used to run multiple instances of Isabelle for concurrent checking.

**Machine learning for automated theorem proving**  Recent efforts have integrated large language models with theorem proving (Jiang et al., 2021, 2022b; Wang et al., 2023). Such techniques build capable LLMs as black-box distribution generators that suggest proof steps or whole proofs, which are then verified by an interactive proof system. The first step is to build a paradigm that captures the task of theorem proving in the context of language modeling formulated as a triple $(\mathcal{A}, S, T)$ where $A \subset \Sigma^*$ is the set of proof steps, $S$ is the set of proof states, $T : S \times A \to S$ is the state transition function which applies proof steps to states (Liu et al., 2025). To use LLMs as black boxes or capable proof assistants (Agrawal et al., 2022), when provided with a theorem statement $x_f \in \Sigma^*$, it needs to provide a valid proof $y_f \in \Sigma^*$ which is valid if applying the proof steps results in a terminal state where $s_{x_f} \parallel y_f$.finish = True.

**Neural theorem proving**  Neural theorem provers combine neural language models (LLMs) with symbolic proof assistants to address formal mathematical tasks. Early implementations focused on premise selection (Irving et al., 2016) which have been shown to be highly effective in guiding proof searches (Wang et al., 2017). Proof search strategies (Polu and Sutskever, 2020) explore the space of possible proofs by generating intermediate steps or tactics. Two primary methodologies for neural theorem proving have emerged: single-pass methods and proof-step methods. Single-pass methods such as DSP (Jiang et al., 2022a), LEGO-PROVER (Wang et al., 2023), and Lyra (Zheng et al., 2023) aim to generate entire proofs at once using prompts enriched with contextual information. Proof-step methods decompose the proving process into incremental steps. These methods, such as GPT-f (Polu and Sutskever, 2020) and POETRY (Wang et al., 2025) utilize LLMs to generate individual tactics or proof steps conditioned on the current state of the proof environment.

**Reinforcement learning for theorem proving**  Reinforcement Learning (RL) for theorem proving emphasizes model learning through direct feedback via trial and error. Early approaches utilizing RL had been unsuccessful due to the infinite action space as well as the absence of a direct self-play setup (Polu et al., 2022). Recently RL approaches have been met with success with the introduction of powerful training regimes like Direct Preference Optimization (DPO) (Rafailov et al., 2023), Proximal Policy Optimization (PPO) (Schulman et al., 2017), and GRPO (Shao et al., 2024). RL frameworks for theorem proving model interactions between LLMs and generated proofs as a Markov Decision Process (MDP). The reward function is typically designed around binary proof completion: assigning a reward of 1 if the proof is verified as correct and 0 otherwise (Dong et al., 2024). This binary reward system provides clear feedback for optimizing performance. Other approaches incorporate search algorithms into RL frameworks, such as Best-First Search (BFS) which guides proof generation by prioritizing promising paths based on heuristic evaluations (Yang et al., 2023); and Monte Carlo Tree Search (MCTS) which explores potential proof paths systematically by balancing exploration and exploitation (Lample et al., 2022). RL-based approaches aim to improve both single-pass and stepwise proof generation models. In single-pass methods, RL optimizes full-proof generation by rewarding logical consistency using Chain-of-Thought tokens and verification success. In stepwise methods, RL enhances tactic prediction by refining intermediate steps based on feedback from symbolic verifiers. Recent works such as DeepSeekMath (Xin et al., 2024) demonstrate that reinforcement learning can significantly enhance models' reasoning abilities by improving their capacity to generate coherent and verifiable proofs over time.

## 3. Method

Building on the work of the workflow described in DSP (Jiang et al., 2022a) and the proof construction method in ProofAug (Liu et al., 2025), in this section we present our framework

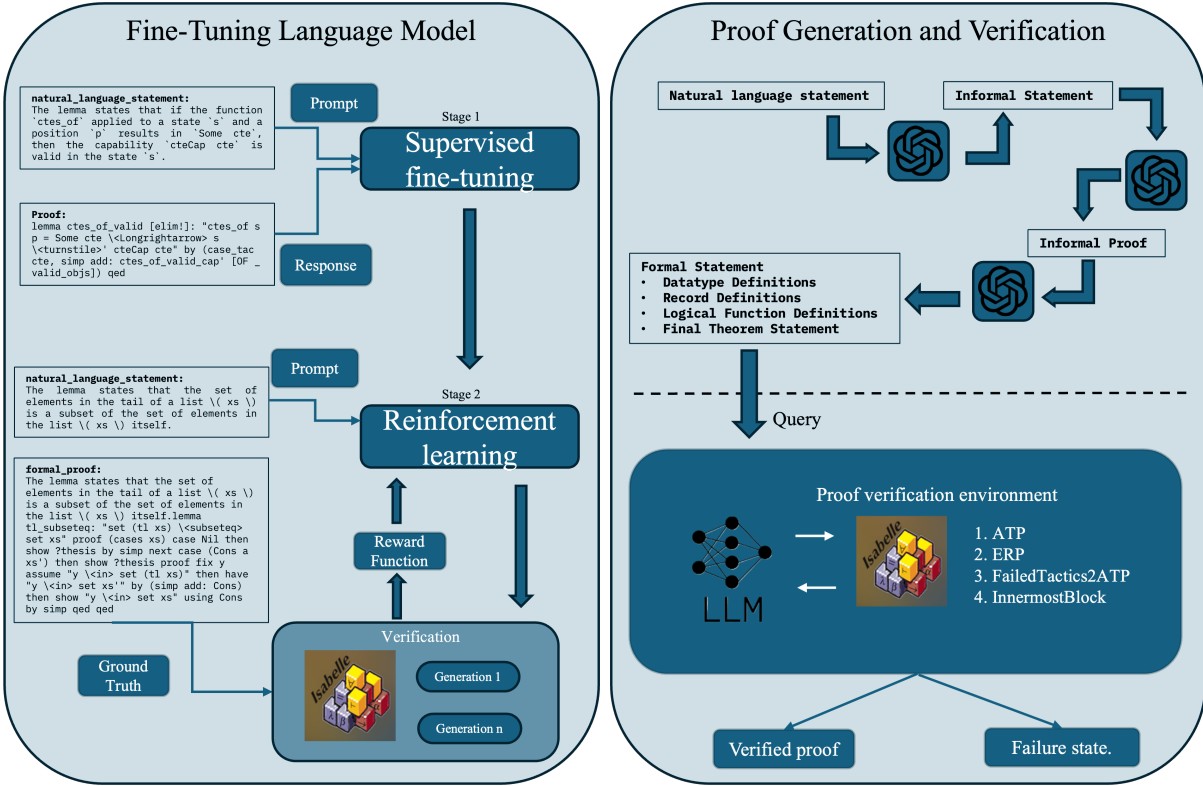

Figure 1: The two core components within the PROOFSEEK framework: (a) the fine-tuning language model module, (b) the proof generation and verification module

PROOFSEEK that leverages the strengths of proof-step and whole-proof generation, as well as the natural language generation paradigms of LLMs. PROOFSEEK consists of two core components: a component for fine-tuning a language model using SFT and RL, and a proof generation and verification component for generating and building the formal proofs. Both components of PROOFSEEK are shown in Figure 1. **Our framework is generalizable across domains where the input is a mathematical statement, policy code, or natural language statement, and the output is a verified proof state or a failure state**. The details of PROOFSEEK is shown in Algorithm 1. We first fine-tune a whole-proof generation model using our two-stage approach. Then, we build a formal statement that represents the provided policy code or mathematical statement. Finally, we leverage the fine-grained proof structure analysis method of ProofAug to verify the generated formal statements.

## 3.1. Fine-Tuning Language Model

We first discuss how we fine-tune both the supervised and reinforcement learning stages of our approach (Figure 1(a), Algorithm 1 lines 1-2). For our training task, we opt to fine-tune our model for whole-proof generation, treating the construction of formal proofs as a general code completion task. The model aims to generate entire proof code based on a theorem in a single step. This scalable approach has been shown to be effective for both training and inference deployment (Xin et al., 2024). Several recent efforts have explored training models specifically for theorem proving, including LLEMMA, which was pretrained using Code Llama on Proof-Pile-2 and outperformed other open-source models on benchmarks such as MATH and Minerva (Lewkowycz et al., 2022). Given the availability of more capable open-source models, we select DeepSeek-Math-7B-rl (Shao et al., 2024) as our base model due to its superior performance in mathematical reasoning tasks. Our two-stage fine-tuning process consists of Supervised Fine-

Tuning (SFT) on curated theorem-proof pairs, followed by Reinforcement Learning (RL) Fine-Tuning using correctness-based reward signals. By integrating formal proof checking into the reward mechanism, we enhance the model's ability to generate valid and verifiable proofs.

### 3.1.1. DATASET CONSTRUCTION

For the two separate stages of fine-tuning, we build different datasets for each purpose. To ensure there are no data leaks during training (i.e., the RL training already sees a particular statement-proof pair), we filter the dataset using a PISA setup. We start with $FVEL_{ER}$, which includes verification tasks formulated in Isabelle, containing 758 theories, 29,125 lemmas, and 200,646 proof steps in total, with in-depth dependencies (Lin* et al., 2024). This translates to 26,192 statements accompanied by their proofs and proof steps. As noted by Lin* et al. (2024), the proofs contain complex dependencies. Due to the setup of the RL training environment, the dataset we need requires "self-contained" proofs or complete proofs that return a verified proof state without further processing. To accommodate this, we filter $FVEL_{ER}$ using PISA to retain only the proofs that return true when iterated through the dataset. The other proofs, while correct, require additional dependencies. The filtered dataset, containing 1,138 statement-proof pairs, was used for the RL stage.

Meanwhile, the remaining 25,054 pairs were allocated for the supervised fine-tuning stage. Due to the nature of our intended application—generalizability for proof generation across domains—we needed to curate a dataset for instruction tuning that included a natural language description of the statement to be proved and its corresponding proofs. We processed this dataset using OpenAI API calls (GPT-4o) to build an SFT dataset in the form (*proof*, *statement*, *natural_language_statement*). For this iteration, we selected 2,000 samples from the 25,054 unproven statements for API prompting. After this process, we arrived at 1,981 samples for the SFT stage and, in a similar fashion, 1,138 samples in the form (*natural_language_statement*, *formal_proof*) for the RL stage.

### 3.1.2. SUPERVISED FINE-TUNING

We first fine-tune a large language model to generate formal proofs in the Isabelle theorem prover. We leverage Unsloth's optimized training framework for parameter-efficient fine-tuning (PEFT). The model is trained using instruction tuning, where each training instance consists of theorem statements paired with corresponding Isabelle proofs to ensure structured learning. We use the FastLanguageModel library from Unsloth, which supports high-efficiency fine-tuning and inference. Instead of fine-tuning the full model, we use Low-Rank Adaptation (LoRA) (Hu et al., 2022) to fine-tune the relevant portions, significantly reducing computational overhead while maintaining high performance. We initialize DeepSeek-Math-7B-rl as the base model. We fine-tune our model using SFTTrainer. After training, the fine-tuned model is published to the Hugging Face Model Hub for further tuning.

### 3.1.3. REINFORCEMENT LEARNING

In the second stage of our fine-tuning approach, we employ Group Relative Policy Optimization (GRPO) (Shao et al., 2024) as our reinforcement learning (RL) algorithm, which has demonstrated superior effectiveness and efficiency compared to Proximal Policy Optimization (PPO) (Schulman et al., 2017). Unlike PPO, which updates a policy based on absolute reward values, GRPO optimizes model updates by considering the relative ranking of multiple generated outputs, making it well-suited for structured text generation tasks.

GRPO operates by sampling multiple candidate proofs for each theorem prompt and optimizing the model based on relative rewards assigned to outputs within the group. This method improves training stability and encourages the generation of higher-quality proofs by leveraging

---

**Algorithm 1** PROOFSEEK

---

**Require:** Natural language statement $s$, fine-tuned whole-proof generation model $\pi_\theta$, proof verification environment.

    **Fine-Tune Model:** Train a whole-proof generation model $\pi_\theta$ using a two-stage approach.

1:    $\pi_\theta = \arg\max_\theta \mathbb{E}_{(x,y)\sim\mathcal{D}}[\log P_\theta(y|x)]$

2:    where $\mathcal{D}$ is the dataset containing natural language statements and their proofs.

    **Autoformalization:** Use a structured workflow to construct a formal statement $S$ from the provided code or mathematical statement by generating:

3:    Datatype definitions $\mathcal{D} = \{D_i\}$ capturing entities involved.

4:    Record definitions $R = \{r_j\}$ for structured objects (e.g., access policies).

5:    Logical function definitions $F = \{f_k : D_i \to D_j\}$.

6:    The final theorem statement as $\forall x \in X, P(x) \Rightarrow Q(x)$.

    **Proof Construction:** Apply ProofAug for verification

7:    Model proof generation as a state transition system $(\mathcal{A}, S, T)$:

8:      $\mathcal{A}$: Finite set of proof actions (inference steps).

9:      $S$: Finite set of proof states.

10:    $T : S \times \mathcal{A} \to S$: Transition function under inference rules.

11:    Generate proof steps iteratively using $\pi_\theta$

12:    $a^* = \arg\max_{a\in\mathcal{A}} P(a|S)$, during inference

13:    Validate each step $a[i]$ using ATP:

14:    $error \leftarrow T(s_{\text{this}}, \langle ATP \rangle).error$

15:    If ATP fails, apply ERP correction:

16:    $y'_f \sim \pi(p(x_i||y_i, x_f||y_f))$

17:    If ERP fails, apply heuristic tactics:

18:    $y'_f \leftarrow \textsc{FailedTactics2ATP}(y_f)$

19:    If no valid step exists, backtrack to the last valid proof block:

20:    $block \leftarrow \textsc{InnermostBlock}(i, a)$

21:    Terminate when a valid proof state or failure state $\emptyset$ is reached.

22: **return** Verified proof or failure state.

---

pairwise ranking rather than relying solely on absolute correctness metrics. The reward function is designed to interact with PISA to verify the generated proofs.

We implement GRPO using Unsloth's FastLanguageModel (similar to the SFT stage), for high-efficiency training with parameter-efficient fine-tuning (PEFT) using Low-Rank Adaptation (LoRA). To enhance proof validity and structure, we employ two reward functions:

1. Correctness Reward: Extracts the Isabelle proof from the model's response and compares it to the ground truth proof.

2. Formal Proof Verification via PISA: Utilizes PISA for proof checking. If a generated proof verifies in Isabelle, it receives a reward of 1; otherwise, 0.

Finally, the trained model is uploaded to the Hugging Face Model Hub for inference[2].

## 3.2. Autoformalization

The first stage of verification in PROOFSEEK involves the autoformalization content being verified (Algorithm 1 lines 3-6). We employ semantic parsing to translate a natural language statement into its logical form. We do this by generating intermediary stages, turning the natural language statement into an informal statement, then deriving an informal proof and a formal statement that is a representation of the initial statement in its logical form for the prover.

---

2. The model can be downloaded from: https://huggingface.co/kings-crown/ProofSeek_v1

### 3.2.1. INFORMAL REPRESENTATION

We start with an informal dataset where $\mathcal{N} = (s_i^{\mathcal{N}})_{i=1}^{|\mathcal{N}|}$ represents natural language statements. For each $s_i^{\mathcal{N}}$, we prompt an LLM (we use GPT-4o) to generate an informal description $d_i$ to provide additional interpretability and structure for both the user and the model in the next steps. Given $(s_i^{\mathcal{N}}, d_i)$, the LLM then produces an informal proof $p_i$ in the same format through curated prompts. This informal proof $p_i$ is seen as the skeleton for the formal representation.

### 3.2.2. FORMAL PROOF REPRESENTATION

To construct a valid Isabelle/HOL proof, the logical form must accurately reflect the original statement. The transformation from natural language to formal representation follows a structured pipeline, leveraging incremental representation generations (which Jiang et al. (2022a) showed to be more efficient than single-shot representation generation). The generated informal proof $p_i$ allows the generated formalization $S$ to be more consistent and faithful. Our approach ensures that the generated proof includes:

- **Datatype Definitions** ($\mathcal{D} = \{D_i\}$): Define structured entities present in the statement. These serve as the foundational building blocks for formal reasoning.

- **Record Definitions** ($R = \{r_j\}$): Represent structured objects, such as access policies or logical relations. Used to define attributes and relationships between entities.

- **Logical Function Definitions** ($F = \{f_k : D_i \rightarrow D_j\}$): Encode logical operations and transformations essential for proof construction. Define predicates and functions that express constraints and properties.

- **Final Theorem Statement** ($\forall x \in X, P(x) \Rightarrow Q(x)$): The theorem statement that encapsulates the key property to be proved. This formally expresses the intended logical relationship in a structured way.

We employ stepwise prompting to sequentially construct the theorem statement:

1. **Natural Language Input** ($s$): The initial informal statement describing what should be proved.

2. **Informal Description** ($d$): A structured interpretation that clarifies the semantics of $s$.

3. **Informal Proof** ($p$): High-level reasoning outlining requirements to guide formalization.

4. **Formal Statement** ($S$): A logically rigorous theorem statement, translation of $p$ into theorem-prover-compatible syntax.

## 3.3. Proof Construction

Once the formal statement $S$ is generated, we employ an interactive theorem prover (ITP)-based approach, using prompting inspired by Jiang et al. (2022a), to guide proof-step generation and construction (Algorithm 1 lines 7-22). We use ProofAug (Liu et al., 2025), an augmentation strategy that refines proofs by integrating automated theorem proving (ATP), efficient recursive proving (ERP), and heuristic-based tactic generation. The proof construction process in ProofAug that we use follows a structured pipeline, which we summarize as follows:

1. **Proof representation in ITP:** Model the proof as a state transition system $(\mathcal{A}, S, T)$.

2. **Proof-step generation via language models:** A fine-tuned generative language model $\pi_\theta$ predicts valid proof steps $a \in \mathcal{A}$ conditioned on the proof state:

$$\pi_\theta(x) = \arg\max_{a \in \mathcal{A}} P(a|S)$$

3. **ATP validation:** Each generated proof step is validated using ATPs. If a proof step is not trivially correct and has *sorry* proofs the system invokes an ATP-based evaluation:

$$error \leftarrow T(s_{\text{this}}, \langle ATP \rangle).error$$

4. **Efficient recursive proving via ERP module:** If ATP validation fails, ERP attempts an alternative inference:

$$y_f' \sim \pi(p(x_i || y_i, x_f || y_f))$$

where $y_f'$ is a corrected proof step. If ERP succeeds, the proof step is updated accordingly.

5. **Heuristic-based tactics for failed proofs:** When both ATP and ERP modules fail, heuristic-based fallback strategies are applied via:

$$y_f' \leftarrow \text{FAILEDTACTICS2ATP}(y_f)$$

attempting to construct missing proof steps via structured heuristics.

6. **Backtracking and Proof Reorganization:** If no valid proof step can be generated, the algorithm identifies the innermost valid proof block, resets proof state, and re-attempts construction:

$$block \leftarrow \text{INNERMOSTBLOCK}(i, a)$$

Given a theorem statement $x_f$, the system iteratively applies proof steps $a \in \mathcal{A}$ until the proof reaches a valid terminal state, yielding either: a **successful proof** $(P)$, where all proof steps are verified; or a **failure state** $(\emptyset)$, if the proof cannot be completed.

## 4. Experiments

To evaluate the utility of our PROOFSEEK framework and assess any enhancements to the model's theorem proving capability, we aim to answer the following research questions:
**RQ1:** How effective is the PROOFSEEK framework at autoformalization and generating proofs in an unseen problem domain?
**RQ2:** How effective is the fine-tuning approach within PROOFSEEK in enhancing the theorem proving capabilities of a language model?

### 4.1. Experiment Setup

To fine-tune our models and evaluate our framework, we utilize two machines (for running concurrent processes). We set up a PISA environment (Jiang et al., 2021) to interact with Isabelle 2022. Our $\langle ATP \rangle$ method uses 8 Isabelle proof methods (auto, simp, auto, blast, fastforce, eval, sos, arith, simp:field simps, simp add:mod simps) as well as Sledgehammer. For the verification process, we use 4 instances of PISA and similar to Jiang et al. (2022a) we set set the timeout for any proof step and Sledgehammer as 10 seconds and 40 seconds, respectively. We run our experiments on: AMD EPYC 7763 64-Core Processor CPU @ 2.49GHz with a NVIDIA A40-48Q and an AMD Ryzen Threadripper PRO 5975WX 32-Core Processor CPU @ 7.00GHz with 2 NVIDIA RTX A6000.

To evaluate our framework, we use two language models for comparison: (1) Deepseek-Math-7b-base (as the base-case) to reproduce the results of ProofAug (Liu et al., 2025); and (2) our fine-tuned model (fine-tuned using the PROOFSEEK framework) to evaluate improvements we can achieve using their method. We follow an approach similar to Liu et al. (2025) and opt to use 1-shot prompting. To keep the evaluation consistent, we limit the sample budget across all the tests to 10, and consistently use a sampling temperature $T = 0.6$ with top $p = 0.95$.

### 4.1.1. BENCHMARK ON MINIF2F-TEST

We evaluate our approach using the miniF2F-Test dataset(Zheng et al., 2021), which includes 488 formal mathematical problems which encompasses high-school level exercises and competition problems. The dataset is split into a validation set and a test set, each containing 244 problems (Xin et al., 2024). We use the Isabelle part of the miniF2F-test dataset that contains an additional informal statement and informal draft for each problem (Jiang et al., 2022a).

### 4.1.2. CASE STUDY: VERIFYING CORRECTNESS OF AWS S3 BUCKET POLICIES

Amazon Web Services (AWS) allows users to create access control policies for managing access to AWS services. These policies regulate access through declarative statements that specify whether a given access control request should be allowed or denied. Given an access control request and an associated policy, access is granted *if and only if* there exists at least one statement in the policy that allows the access and no statement that explicitly denies it. Thus, to verify the correctness of a policy, one must reason about the logic of the policy statements and determine whether the intended behavior (as expressed in natural language) matches the formal semantics of the policy. If the intent aligns with the semantics, the policy is considered correct with respect to that intention. Otherwise it is incorrect. For full details of the AWS policy language, we refer the reader to Backes et al. (2018).

In this case study, we consider a set of S3 bucket policies along with a set of generated natural language description of their intended behaviors. We evaluate the effectiveness of our approach by generating Isabelle proofs that demonstrate the correctness of the policies with respect to their intended access control intents. We consider two scenarios:

**Scenario 1: Evaluation with manually curated dataset** For the first phase of our analysis, we investigate the proving ability of our setup using a small, manually curated dataset of S3 bucket access policies. We randomly pick 25 policies from the Quacky dataset (Eiers et al., 2022) and construct formal statements for each policy code in the proof environment. We extract key components from the policy (Actions, Resources, Effects, Conditions) and convert them into formal logic types. Then, we finally transform the policy components into formal entries for Isabelle. The complete evaluation comprises 25 policy-formal statement pairs.

**Scenario 2: Evaluation on LLM generated dataset** To evaluate the utility of the framework as a whole, we automate the formalization process of a set of AWS S3 Bucket Policies in a csv file using GPT-4o to generate formal statements to be proved. We use an approach inspired by DSP (Jiang et al., 2022a) to iteratively prompt the LLM to autoformalize the policy code as described in our framework. We use some of the manually curated policy-formal statement pairs as few-shot examples to guide the model along. The final dataset comprises of 243 policies, and we use the method described in ProofAug with our model to prove the statements.

### 4.2. Experiment Results

Here we present the results of our experiments on different datasets, including the miniF2F-Test dataset for mathematical theorem proving and the Quacky dataset for AWS S3 bucket policy verification[3]. Our evaluation primarily focuses on the proof success rate, the average number of attempts required, and the total execution time.

Table 1 reports the results on the miniF2F-Test dataset, comparing our approach, PROOF-SEEK, against DeepSeek. We evaluate both models with and without efficient recursive proving (ERP). PROOFSEEK achieves a 40.1% success rate without ERP, slightly lower than DeepSeek (41.8%). The total execution time for ProofSeek (ERP: 38,651.78s, No ERP: 36,557.68s) is slightly lower than DeepSeek with ERP (53,658.5s).

For manually curated AWS S3 bucket policies from the Quacky dataset, PROOFSEEK achieves a 96.0% success rate across all settings. This is unsurprising as the proof formulations were of

---

3. The generated proof construction jsonl files are attached in the code repository

Table 1: Experimental Results on Evaluation Datasets

| Method | Success Rate (%) | Avg Attempts | Total Exec. Time (h:mm:ss) |
|---|---|---|---|
| **MiniF2F Dataset (245 Problems)** | | | |
| ProofSeek (Ours) (ERP) | 41.8 | 5.66 | 10:44:11 |
| ProofSeek (Ours) (No ERP) | 40.1 | 5.67 | 10:09:17 |
| DeepSeek (ERP) | 42.2 | 5.59 | 14:54:18 |
| DeepSeek (No ERP) | 41.8 | 5.54 | 07:24:12 |
| **Curated Quacky Dataset (25 Problems)** | | | |
| ProofSeek (Ours) (ERP) | 96.0 | 0.60 | 00:06:33 |
| ProofSeek (Ours) (No ERP) | 96.0 | 0.44 | 00:03:16 |
| DeepSeek (ERP) | 96.0 | 0.64 | 00:09:33 |
| DeepSeek (No ERP) | 96.0 | 0.84 | 00:06:15 |
| **Generated Quacky Dataset (243 Problems)** | | | |
| ProofSeek (Ours) (ERP) | 66.6 | 1.33 | 00:20:34 |
| ProofSeek (Ours) (No ERP) | **69.1** | 1.15 | **00:20:27** |
| DeepSeek (ERP) | 63.3 | 2.05 | 00:26:52 |
| DeepSeek (No ERP) | 66.6 | 1.95 | 00:24:36 |

high quality making it easy for the prover. This is also reflected in the times: ProofSeek (No ERP) is the fastest, completing in 196.01s, compared to DeepSeek (ERP: 573.84s, No ERP: 375.67s), demonstrating superior efficiency. When evaluating ProofSeek on LLM-generated policy statements: ProofSeek (No ERP) outperforms all other settings with a 69.1% success rate, surpassing DeepSeek.

**MiniF2F-Test Performance (RQ1)**: Our framework demonstrates effectiveness in auto-formalization and proof generation in unseen domains, achieving performance comparable to DeepSeek while improving computational efficiency.

**AWS S3 Policy Verification (RQ2)**: Our system's fine-tuning results in highly efficient and accurate verification of structured policies, confirming enhancements in theorem proving capabilities. Moreover, ProofSeek proves more effective on LLM-generated formalizations, demonstrating robustness in handling formalization related tasks in generalized problem domains and maintaining higher success rates over DeepSeek.

## 5. Discussion and Future Work

Although our framework demonstrated its practical utility for real-world use cases, our results on benchmark datasets remain behind SOTA approaches. Additionally, we speculate that the fine-tuning process enhanced the theorem-proving capability of the language model, as evidenced by the smaller number of proof attempts required to complete proofs. However, due to the construction of the symbolic methods being used, it did not perform as well on the benchmark as expected. Moreover, from our initial results, it is clear that further fine-tuning is necessary, both using supervised and RL-based methods. In this work, we refrained from further training so as to not overfit. We also look forward to incorporating reasoning-based feedback (Xie et al., 2025) to build better models for formal verification.

We believe that an important direction for future work is to fully leverage the reliability aspect of language model-generated proofs across systems. Reliability is a highly desirable property in inherently probabilistic systems. In the future, we aim to incorporate other forms of symbolic systems, such as knowledge graphs, to make automated theorem proving with LLMs more reliable and consistent.

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

## Appendix A.

LLM generated formalization for access policy code. This is a representative example that takes the reader through the different stages of the Proof Generation and verification as show in Figure 1.

Listing 1: Problem Name

```
{
  "problem_name": "s3_samples_mutations_ec2_exp_single_ec2
      _prevent_running_classic_policy_6_0"
}
```

Listing 2: EC2 Access Policy JSON

```
{
  "policy_json": {
    "Statement": [
      {
        "Effect": "Allow",
        "Action": "ec2:RunInstances",
```

```
 7          "Resource": "arn:aws:ec2:us-east-1:123412341234:*"
 8        },
 9        {
10          "Effect": "Allow",
11          "Action": "ec2:RunInstances",
12          "Resource": [
13            "arn:aws:ec2:us-east-1::image/ami-*",
14            "arn:aws:ec2:us-east-1:123412341243:instance/*",
15            "arn:aws:ec2:us-east-1:123412341234:volume/*",
16            "arn:aws:ec2:us-east-1:123412341234:network-interface/*",
17            "arn:aws:ec2:us-east-1:123412341234:key-pair/*"
18          ]
19        }
20      ]
21    }
22 }
```

#### Listing 3: Informal Statement

```
 1 {
 2   "informal_statement": "The text you provided is a policy statement written in
       JSON format, which is typically used in cloud computing environments like
       Amazon Web Services (AWS) to define permissions. Here's a breakdown of
       what it means in plain English:\n\n1. General Permission:\n - The policy
       allows the action \"ec2:RunInstances.\" This means that the user or
       service with this policy can start or launch new EC2 instances.\n - This
       applies to any resource within the specified AWS account (123412341234) in
        the \"us-east-1\" region.\n\n2. Specific Permissions:\n - Images: use any
        AMI in \"us-east-1\".\n - Instances: manage EC2 instances in 123412341243
       .\n - Volumes: manage EBS volumes in 123412341234.\n - Network Interfaces
       and Key Pairs: full control in 123412341234.\n\nSummary: This policy
       allows launching and managing EC2 instances and their dependencies."
 3 }
```

#### Listing 4: Informal Proof

```
 1 {
 2   "informal_proof": "To provide an informal proof or argument supporting the
       interpretation of the JSON policy statement, we break it into structured
       observations:\n\n1. JSON uses keys like \"Effect\", \"Action\", and \"
       Resource\" to structure permissions.\n2. 'Allow' means access is granted;
       'ec2:RunInstances' lets the user launch EC2s.\n3. '*' in the ARN means it
       applies to all of that type within the account and region.\n4. Specific
       ARNs grant permissions to manage AMIs, Instances, Volumes, Interfaces, and
        Key Pairs.\n\nConclusion: These combined statements demonstrate full EC2
       launch and management capability."
 3 }
```

#### Listing 5: Formal Statement

```
 1 datatype ec2_action = RunInstances
 2
 3 datatype ec2_resource = AllResources | Images | Instances | Volumes |
     NetworkInterfaces | KeyPairs
 4
 5 datatype principal = Anyone
 6
 7 record policy_entry =
 8   act :: ec2_action
 9   res :: ec2_resource
10   prin :: principal
11
```

```
12 definition ec2_instance_policy :: policy_entry where
13   "ec2_instance_policy = (|
14     act = RunInstances,
15     res = AllResources,
16     prin = Anyone
17   |)"
18
19 fun policy_allows :: "policy_entry => ec2_action => ec2_resource => bool" where
20   "policy_allows pe a r = (act pe = RunInstances ∧ (res pe = AllResources
         res pe = r))"
21
22 theorem ec2_policy_correctness:
23   shows "policy_allows ec2_instance_policy RunInstances AllResources ∧
24           policy_allows ec2_instance_policy RunInstances Images ∧
25           policy_allows ec2_instance_policy RunInstances Instances ∧
26           policy_allows ec2_instance_policy RunInstances Volumes ∧
27           policy_allows ec2_instance_policy RunInstances NetworkInterfaces ∧
28           policy_allows ec2_instance_policy RunInstances KeyPairs"
29   oops
```

### Listing 6: Initial Proof Attempt

```
1 (* Proof of the theorem *)
2 (*
3 proof -
4   have "policy_allows ec2_instance_policy RunInstances AllResources"
5     by (simp add: ec2_instance_policy_def)
6   moreover have "policy_allows ec2_instance_policy RunInstances Images"
7     by (simp add: ec2_instance_policy_def)
8   moreover have "policy_allows ec2_instance_policy RunInstances Instances"
9     by (simp add: ec2_instance_policy_def)
10   moreover have "policy_allows ec2_instance_policy RunInstances Volumes"
11     by (simp add: ec2_instance_policy_def)
12   moreover have "policy_allows ec2_instance_policy RunInstances
         NetworkInterfaces"
13     by (simp add: ec2_instance_policy_def)
14   moreover have "policy_allows ec2_instance_policy RunInstances KeyPairs"
15     by (simp add: ec2_instance_policy_def)
16   ultimately show ?thesis by simp
17 qed
18 *)
```

### Listing 7: Sorry Proof (Commented)

```
1 (* Proof of the theorem *)
2 (*
3 proof -
4   have "policy_allows ec2_instance_policy RunInstances AllResources"
5     by (simp add: ec2_instance_policy_def)
6   moreover have "policy_allows ec2_instance_policy RunInstances Images"
7     by (simp add: ec2_instance_policy_def)
8   moreover have "policy_allows ec2_instance_policy RunInstances Instances"
9     by (simp add: ec2_instance_policy_def)
10   moreover have "policy_allows ec2_instance_policy RunInstances Volumes"
11     by (simp add: ec2_instance_policy_def)
12   moreover have "policy_allows ec2_instance_policy RunInstances
         NetworkInterfaces"
13     by (simp add: ec2_instance_policy_def)
14   moreover have "policy_allows ec2_instance_policy RunInstances KeyPairs"
15     by (simp add: ec2_instance_policy_def)
16   ultimately show ?thesis by simp
17 qed
```

```
18  *)
```

Listing 8: State Information

```
1  {
2    "success": true,
3    "i_try": 0,
4    "success_stage": "init_proof",
5    "has_timeout": false,
6    "extra_calls": 0,
7    "has_sc": false
8  }
```

