# OpenReview forum: "Neural Theorem Proving: Generating and Structuring Proofs for Formal Verification"
_nesyconf.org/NeSy/2025/Conference — NeSy 2025 Poster_

### Official Review · Reviewer_Ur9A · 2025-03-24
**Neural generation of proofs for Isabelle which unfortunately doesn't improve over DeepSeek**

**Rating:** 5
**Confidence:** 4

**Review:**

The paper presents ProofSeek, an LLM-based approach for producing proofs for the Isabelle proof assistant.  The methodology builds on a previous method called ProofAug, which appear to be the current state of the art on the MiniF2F dataset (according to paperswithcode.com).

Weakness:
-----------------------------
The experimental results are somewhat disappointing, in that ProofSeek doesn't improve over a plain DeepSeek model on miniF2F, nor on a curated dataset. It shows a small improvement over DeepSeek on one synthetic dataset, but I must say I didn't quite understand exactly what the characteristics of this dataset were, and how it differed from the hand curated set.

 Results aside, I'm also feeling that there are important implementational details missing. For instance, the work seem to build upon ProofAug, but things like ERP (section 3.3) is never properly explained or introduced. Neither are any details on the "heuristic fallback" provided. So it is difficult to get a proper grip of the methodology, and the paper would benefit from being a bit more self contained.

In the conclusion, the authors seem to suggest they need to do longer fine-tuning for reading better results. I suggest they try that and submit the paper again once this has been tried. Perhaps you will have more interesting results then. I would also encourage the authors to include the most recent state of the art model, which appear to be ProofAug, which you use as part of your system, in the evaluation.

Additional comments:
-----------------------------

p. 3: last sentence in first paragraph: very long and difficult to read. Consider rephrasing and clarifying.

p.3: In fact, both the first and second paragraph seem to talk about roughly the same things. Why are there two separate paragraphs? Both talk for instance about interactive theorem provers and learning for them. Consider rearranging the text for better clarity and flow.

section 3.1.1: "we need requires “self-contained” proofs or complete proofs " ---> I didn't understand what you mean by "self-contained" va complete proofs here. What is the former even? What's the difference between the two?

p. 10: typo: "high quality making it est for the prover."

Table 1, page 10: Consider writing the timings not in seconds but maybe min:sec (or even h:min:sec) as it's difficult to get a direct feel for how long 30000-ish seconds really is.

**Anonymity:**

Remain anonymous

---

### Official Review · Reviewer_QmfT · 2025-03-29
**Too fast for me**

**Rating:** 6
**Confidence:** 2

**Review:**

The paper says it is about formal verification of code using LLMs. It seems to be about "end to end" LLM-enabled solving of puzzles
written in natural language, which could come from software verification or from something else. They generate a formal statement from natural language, then use an interactive theorem prover to find a proof, and then verify it using a theorem-proving. The paper describes the system and includes an experimental analysis on access control policies

In terms of how they do it, I personally could not follow it. But this could well be because of my lack of background. I know automated theorem proving fairly well, and have a bit of background in RL and interactive theorem proving. But I known nothing about LLMs, fine-tuned or otherwise.  And at this point (as the authors point out) there is a fair amount of work in using LLMs with theorem provers, and I am not aware of that. Another factor could be that the authors had to omit many details to fit the page limit. Finally, I acknowledge that a) systems like the one described here, which integrate a lot of pre-existing components from distinct and highly technical areas, are hard to describe in a self-contained manner; and b) for a newish conference like NeSy it is hard to know what background to assume.

Overall, this might be a good or even very good contribution to NeSy -- but I cannot confirm that.

Just to confess further, I could follow the diagram on page 4 -- the system consists of fine tuning, and then some kind of interaction with an ITP. But other than that I was pretty much lost.

In some cases there were references to things I was unfamiliar with, but which are likely standard. Here are (just a few) examples:

"using a PISA setup"

"We leverage Unsloth’s optimized training framework for parameter-efficient fine-tuning (PEFT)"

top of page 8 "sorry proofs" (page 8: yes, I can imagine what this means, but is this really so well-known?)

In other cases, I was not sure that the references were standard. An example is Efficient Recursive Proving (ERP): in general, and in particular within line 15 of algorithm 1, which is further described on page 8: "ERP attempts an alternative inference .. (*)... where.... is a corrected proof step.

what is a corrected proof step? And what does the equation that I have abbreviated above as (*) really mean in the context of the algorithm?

A number of steps in the algorithm were extremely high-level: e.g. "attempting to construct missing proof steps via structured heuristics."

There are other terms that I know, but I was not sure how they were applied.
For example, in fine tuning there are RL and rewards -- fine. But then "GRPO operates by sampling multiple candidate proofs for each theorem prompt and optimizing the model based on relative rewards assigned to outputs within the group." So GRPO is a proof-specific method, or is this text just describing how it is applied in the context of the system?

In the description of the base rewards it says the system "Extracts the Isabelle proof from the model’s response and compares it". Maybe this is obvious, but what if the model's response is not a valid Isabelle proof (can't this happen in the fine tuning stage?)

In the experimental section the first example is about "S3 bucket policies". Here I am sure there are access control researchers who know what that means, but would NeSy readers really know it?

**Anonymity:**

Remain anonymous

---

### Official Review · Reviewer_pyB9 · 2025-04-05
**Neural Theorem Proving: Generating and Structuring Proofs for Formal Verification" (ProofSeek) introduces a novel framework leveraging LLMs and heuristics for generalized theorem proving, demonstrating promising results in verifying AWS S3 policies but with benchmark performance slightly behind SOTA approaches.**

**Rating:** 7
**Confidence:** 3

**Review:**

The paper presents a well-defined framework, ProofSeek, for generalized theorem proving. The methodology involves a two-stage fine-tuning process for an LLM (SFT and RL) and the integration of a natural language statement generator and a proof construction module utilizing heuristics from ProofAug. The evaluation includes experiments on the standard miniF2F-test benchmark and a practical case study on verifying AWS S3 bucket policies. The authors compare their approach with a strong baseline (DeepSeek) across different settings (with and without Error Recovery Procedures). While the benchmark results are slightly behind SOTA approaches, the performance on the curated Quacky dataset for AWS S3 policies is promising. The inclusion of execution time as a metric is also valuable.

Clarity:
The paper is generally well-written and structured logically. The introduction clearly outlines the problem and the proposed solution. The "Background and Related Work" section provides relevant context. The "Method" section clearly describes the ProofSeek framework and its components, including the fine-tuning process and proof construction. Figures and Algorithm 1 help in visualizing the framework and the proof generation process. The experimental setup and results are detailed.

Originality:

The main originality lies in the integrated framework ProofSeek, which combines several elements in a novel way. This includes:

• Generating natural language statements of the code or policy to be verified as an initial step.

• Utilizing an LLM for generating whole proofs based on these natural language statements.

• Employing heuristics from ProofAug for building the final proof within a formal language like Isabelle.

• The two-stage fine-tuning process (SFT on syntactically correct Isabelle code followed by RL encouraging verified proofs) is a specific
contribution.

• The curation of a dataset based on the FVELer dataset tailored for their RL training setup is another original aspect.

• The application of the framework to the domain of AWS S3 bucket policy verification showcases its potential beyond standard mathematical benchmarks.

Significance:
The paper addresses a highly relevant and challenging problem: generalized theorem proving for formal verification, especially in the context of LLM-generated code and complex systems like cloud policies. The potential to automate the laborious process of formal verification and to verify the correctness of security policies has significant practical implications. The work contributes to the ongoing efforts in leveraging LLMs for formal reasoning and explores methods to improve their proof generation capabilities through targeted fine-tuning and structured proof building. The framework's generalizability across different input types (code, policies, natural language) is a key strength.

Pros:

• Introduces a novel and comprehensive framework (ProofSeek) for generalized neural theorem proving.

• Effectively combines natural language processing, large language models, and symbolic theorem proving techniques.

• Presents a structured approach to proof generation and verification leveraging existing tools and heuristics.

• Demonstrates the framework's applicability to a real-world problem with the AWS S3 bucket policy verification case study.

• Achieves competitive performance with a strong baseline (DeepSeek) on the miniF2F-test benchmark and shows efficiency gains.

• The two-stage fine-tuning process is designed to enhance both syntactic correctness and semantic validity of generated proofs.

• The curated dataset contributes to the research community for future training tasks in formal theorem proving.

• The framework is designed to be generalizable across different domains beyond mathematics.

Cons:

• Performance on the miniF2F-test benchmark is still slightly lower than the DeepSeek baseline. The authors acknowledge that their results remain behind SOTA approaches on benchmark datasets.

• The paper suggests that the fine-tuning enhanced theorem-proving capability, but the benchmark performance did not fully reflect this, possibly due to the symbolic methods used.

• The authors themselves note the need for further fine-tuning (both SFT and RL-based methods) to achieve better results.

• The complexity of integrating multiple components (LLM, ATP, heuristics) might present engineering challenges for adoption.

**Anonymity:**

Remain anonymous